# Effect of High-Protein and High-Fiber Breaders on Oil Absorption and Quality Attributes in Chicken Nuggets

**DOI:** 10.3390/foods12244463

**Published:** 2023-12-13

**Authors:** Glenda Gutiérrez-Silva, Francisco Vásquez-Lara, Nina G. Heredia-Sandoval, Alma R. Islas-Rubio

**Affiliations:** Coordinación de Tecnología de Alimentos de Origen Vegetal, Centro de Investigación en Alimentación y Desarrollo, A.C. Carretera Gustavo E. Astiazarán Rosas # 46, Colonia La Victoria, Hermosillo 83304, Mexico; glendagtz@gmail.com (G.G.-S.); fvas@ciad.mx (F.V.-L.); nina.heredia@ciad.mx (N.G.H.-S.)

**Keywords:** breaders, chicken nuggets, food frying, nutrition quality, oil absorption

## Abstract

Consumption of fried foods is associated with a higher risk of cardiovascular and other diseases; therefore, consumers are looking to reduce fat intake. We evaluated the effect of high-protein breaders and fiber on oil absorption and quality attributes in chicken nuggets, using flour blends (wheat, chickpea, coconut, oil-quinoa-chia), soy protein concentrate, and brewers’ spent grain. We evaluated the chemical composition, water and oil retention capacity (ORC), viscosity profile, and flour particle size distribution, along with the developed breaders (Formulation 1 and 2) and a commercial breader (CB), in addition to texture, color, fat, and moisture contents of the fried chicken nuggets prepared with the developed breaders and the CB. The total dietary fiber content (TDF) of the nuggets breaded with only Formulation-1 and CB was determined. Nuggets breaded with Formulation-1 showed lower (*p* ≤ 0.05) ORC, better moisture retention (67.6%), and more TDF (4.5% vs. 2.3%, *p* ≤ 0.05) compared to CB-breaded nuggets. Nuggets with Formulation-1 showed the expected texture and color characteristics for fried products. Formulation-1 has the potential to be used as a breader due to its moisture, reduced ORC, and the texture and color it imparts to the fried nuggets, providing higher amounts of nutrients and possible health benefits.

## 1. Introduction

Frying is one of the oldest and most commonly used cooking methods, both industrially and in households. This technique is applicable to a wide variety of foods, including chicken, fish, chips, and vegetables, among others [1,2]. Breaded and fried foods are popular among consumers due to their sensory characteristics, particularly their taste and crispy texture [1,3]. However, these fried products often have a high-fat content, sometimes reaching up to 50% of the total product weight [4]. This high-fat content decreases their nutritional value according to recommendations for a healthy diet, and is associated with the development of cardiovascular diseases, obesity, and diabetes [1,5,6].

Reducing the fat content in fried foods has become a significant challenge, as lipids are responsible for conferring typical taste and texture [4]. The oil absorption process in fried foods is influenced by the composition and viscosity of the oil, frying time, and temperature, as well as the water content. The removal of water from fried foods is affected by the composition, structure, size, shape, and surface type of the food [7,8]. Three mechanisms describe the oil absorption in fried foods: water replacement during frying; the effect in the cooling phase, which describes how the oil is absorbed after frying; and the surfactant theory of oil absorption, which is related to the changes in oil composition to form surfactants [8,9,10].

Several strategies have been proposed to decrease oil absorption in fried foods, including modifying frying techniques, changing the surface dimensions of the food, and using coatings as a barrier against fat. Concerning breaded foods, the composition of their coating becomes crucial in oil absorption during the frying process [7]. The incorporation of hydrocolloids, proteins, cellulose, and their derivatives into breading and batter formulations reduces oil penetration during frying. In 2018, Shan et al. [11] added xanthan gum and soy fiber to a breading formulation, resulting in a 17% reduction in lipid absorption. Similarly, the use of bamboo dietary fiber in fish nuggets reduced oil absorption by 9% [12]. Additionally, using different proteins (wheat, egg, or soy) in a batter for chicken nuggets significantly reduced the fat percentage in the final product, with soy protein isolate being the most effective ingredient [13]. Despite favorable results in reducing oil absorption, essential quality attributes such as texture, color, and flavor have not been evaluated or show unsatisfactory outcomes.

The incorporation of byproducts of the food industry into the development of new food products is a good strategy to enhance the nutritional quality and possible health benefits to the consumer. Coconut flour, a byproduct of the coconut oil industry, has been used to successfully replace (at a 20% level) wheat flour in the preparation of a wheat flour-based tortilla with higher nutritional quality and fiber content [14], and for the production of a baked snack [15]. Another byproduct, rich in insoluble fiber and bioactive compounds, is brewers’ spent grain (BSG), already used to create ingredients for different food matrices. The incorporation of processed BSG-derived ingredients provides a promising approach to improve the nutritional value of pasta while maintaining product quality [16].

Cereal, pulses, and oilseed flours contain proteins and polysaccharides that confer them hydrocolloidal properties. Pulse flours have been used to fortify different foods or partially replace wheat flour. Chickpea flour has been used to produce pasta, biscuits, and other products in combination with wheat flour. Recently, protein-enriched biscuits prepared with chickpea flour and an oat-milk byproduct were developed [17]. The use of functional ingredients such as proteins, non-protein hydrocolloids, and insoluble dietary fibers can be added to the batter to inhibit oil penetration and moisture evaporation during deep-fat frying. Proteins, due to their gel-forming properties at high temperatures, can form networks that act as barriers against water removal and oil penetration [4,7]. Polysaccharides such as starch and fiber have a high water-binding capacity, consequently preventing water removal [4,7]. Despite their ideal characteristics against the oil absorption mechanism, their inclusion in commercial batters and coatings is limited. Therefore, the aim of this study was to develop a breading formulation high in protein and fiber and evaluate its effect on the reduction of oil absorption, texture, and color of breaded and fried chicken nuggets.

## 2. Materials and Methods

### 2.1. Materials

Cooked chickpea flour (CPF) was obtained by cooking chickpeas (*Cicer arietinum*) variety blanco X4 Sinaloa 92 in distilled water (1:3 *w*/*v*) for 50 min. The cooked chickpeas were drained and placed on trays for 1 h at room temperature and then dried in an oven (Model 1321F, VWR, Cornelius, OR, USA) at 60 °C for 24 h. The dried chickpeas were ground into flour in a Pulvex 200 mill (Pulvex de México, S.A. de C.V., CDMX, México) by passing them through a 1 mm sieve. Coconut flour (CF) was prepared according to Islas-Rubio et al. [14]. Oat flakes (*Avena sativa*), quinoa (*Chenopodium quinoa*), and chia (*Salvia hispanica*) seeds, as well as wheat flour (WF), were purchased from the local market. A blend made of 88.0% oat flakes, 8.8% quinoa, and 3.2% chia was put into a pot containing boiling distilled water (4.08 L of water per 1 kg of mixture). Cooking was carried out for 4 min and 45 s with constant stirring. Then, it was removed from the heat and rested for 30 min. Afterward, drying was carried out at 70 °C for 11 h and 45 min in an oven (model VWR 1321F). Then, the dried blend was ground in a Pulvex 200 mill using a 1 mm sieve to constitute oat-quinoa-chia flour (OQC). The moisture content of the raw, cooked, and dry mixture was determined using Method 44-15.01 [18].

Soy protein concentrate (SPC) was supplied by ADM (Protein Specialties Division, Defatur, IL, USA), and brewers’ spent grain (BSG) was prepared by drying wet BSG donated by Cuauhtemoc-Moctezuma-Heineken^®^ (Navojoa, Sonora, México) in an MP-500 dryer (Enviro-Pak, Clackamas, OR, USA) at 40 °C for 40 h. The commercial breader (CB) was bought from a grocery store in Tucson, AZ, USA.

### 2.2. Physicochemical Properties of Flour Samples and Breaders

The proximal analysis of the flours and breaders was carried out according to AACC [18] Methods 44.15.01, 46–13.01, 30.20.01, and 08.01.01 for moisture, protein, fat, and ash, respectively. A conversion factor of 6.25 was used for the protein estimations, except those for CF (5.30), SPC (5.71), WF (5.75), and oat flakes (5.83). Water retention capacity (WRC) and particle size distribution (PSD) were determined according to Methods 56.11.1 and 55-60.1, respectively [18]. Oil retention capacity (ORC) of the flours, breaders (F1 and F2), and CB was determined according to Sarangapani et al. [19].

### 2.3. Viscosity Profile by the Rapid Visco Analyser

Viscosity profiles of the flours and breaders were carried out using a Rapid Visco Analyser (RVA-Super4, Newport Scientific, Warriewood, NSW, Australia) according to Method 76-21 [18]. A 3.5 g sample (14% moisture basis) was directly weighed into an RVA canister, followed by the addition of 25 g of distilled water. The RVA’s STD1 pasting profile was selected, comprising the following steps: initial temperature was set at 50 °C and held for 1 min; then, the sample is heated to 95 °C at a rate of 12 °C/min, maintained at 95 °C for 2 min 30 s, and then cooled to 50 °C at the same rate; lastly, it was held at 50 °C for 2 min. Paddle speed was set at 160 rpm. Parameters recorded were viscosity or peak viscosity, minimum viscosity, drop viscosity, setback, and final viscosity.

### 2.4. Breaders’ Formulations

The formulations of the two breaders are presented in Table 1. The formulations of the breaders (F1 and F2) were established based on the physicochemical characteristics of the individual flours (WF, CF, CPF, OQC, SPC, and BSG). Once the breading formulations were established, they were prepared, and their physicochemical properties were evaluated, as previously mentioned.

### 2.5. Preparation of Unbreaded Chicken Nuggets

Chicken nuggets were prepared using chicken breasts which were ground in a domestic food processor (Model AR6838, Moulinex, CDMX, Mexico) until a paste was formed. To 200 g of the paste, 3 g of salt and 3 g of ground pepper were added and mixed until homogenized. Twenty five grams of the mixture were taken and placed in molds with the following dimensions: 6 cm × 4 cm × 1.5 cm. They were subsequently frozen at −20 °C for 35 min, unmolded, and rolled (4 times each side) until homogenized, removing any excess breading.

### 2.6. Test of the Breaders’ Adhesion to Chicken Nuggets

The percentage of the breaders’ adhesion or “pickup” was determined on the chicken nuggets, which were weighed prior to breading, added to a plastic bag containing 50 g of breading, and stirred lightly for 30 s. Then, the covered nugget was weighed, and the excess breading was removed. The nugget pieces (3 each time) were fried in pure soy edible oil at 180 °C for 50 s each side, set on absorbent paper, and allowed to cool at room temperature for 20 min. The nuggets were weighed after frying [11,20]. The fryer oil was replaced with new oil with each batch of nuggets. Texture profile analysis and color measurements of the breaded and fried chicken nuggets were performed. The pickup percentage was calculated as follows:Pickup% = [(m_1_ − m_2_)/m_1_] × 100,
where m_1_ is the weight of the breaded and fried chicken nugget, and m_2_ is the weight of the unbreaded chicken nugget [21]. The result is expressed as a percentage.

### 2.7. Physicochemical Evaluation of Breaded and Fried Chicken Nuggets

To determine the moisture content, the fried chicken nuggets were ground in a Moulinex food processor (Model AR6838), and 3 g of the ground sample was weighed. The fat was determined from the dry sample according to methods of the AACC [18]. Texture profile analysis was performed using a TA-XT2 texturometer (Texture Analyser, Stable Micro Systems, Surrey, UK), according to Brannan et al. [20] with slight modifications. The sample was cut into 1.5 cm^3^ cubes and compressed 5 mm at a constant speed of 0.5 mm/s with a force of 5 g. The accessory used to perform the compressions was a metal cylinder 3 mm in diameter and 25 mm long (Code TA-53). The textural attributes measured were hardness, springiness, cohesiveness, and chewiness. Color measurement was carried out using a Konica Minolta colorimeter (model Chroma Meter CR-300, Minolta Sensing Inc., Osaka, Japan). The CIE L* (luminosity), a* (variation from red to green), and b* (variation from yellow to blue) values were recorded. Three measurements were performed at different points on each nugget piece. Plastic film was placed over the samples while measurements were taken.

### 2.8. Statistical Analysis

Results from all measurements were presented as mean ± standard deviation. Proximal analysis, PSD, and RVA were determined by triplicates; WRC and ORC were quadruplicates; and FDT by duplicate. Texture and color analyses were the average of 8 and 12 measurements, respectively. Statistical differences between the mean values were analyzed using one-way analysis of variance (ANOVA), followed by the Tuckey’s procedure. NCSS version 07.1.5 (NCSS LLC, Keysville, UT, USA) was used to perform the statistical analysis at the 5% significant level.

## 3. Results and Discussion

### 3.1. Physicochemical Properties of Flour Samples and Breaders

Table 2 shows the proximal composition of each flour used to prepare the breader formulations, F1 and F2 breaders, and CB. All the flours and breaders fulfilled the norm NMX-F-007 regarding the moisture content (a value lower than 14%). The protein values of all the flours were significantly higher than that of the CB, with the SPC standing out with the highest percentage, which meets the protein content specification (80–90%), followed by the BSG and CF. Regarding the ash content, that of the CB was considerably higher; this is attributed to the fact that the CB contains different spices that could increase its value. Finally, WF and SPC obtained fat percentages lower than that of the CB. In general, the data agree with what was described in the literature, with the exception of the protein content of CF, which was higher than that reported by Trinidad et al. [22].

Table 2 also shows the percentages of WRC and ORC of the flour samples, the developed breaders, and the CB. WF and the CB did not present significant differences in WRC, while the rest of the flours had significantly higher values. CPF, despite presenting one of the lowest WRC values, managed to double that of the CB, while CF and SPC obtained values of more than 400% and 350%, respectively. This can be related to the high-protein content present in both flours, which tends to bind water very easily [23]. Also, it has been shown that water absorption increases when the fiber content in food increases; this is because it has a large number of hydroxyl groups with which water can bind [24,25]. Regarding ORC, the lowest values were for WF, CPF, and OQC, which did not present significant differences with the CB (1.82 g/g). The rest of the flours showed values very close to that of the CB, with BSG presenting the highest value of 2.68 g/g. This characteristic is related to particle size, where finer particles tend to absorb less oil, as was the case of WF and the CB [26].

The PSD of the flours and breaders is shown in Figure 1. Flours and the CB showed differences in particle size distribution (Figure 1A). WF was the finest, presenting the greatest percentage at the bottom, passing through the 149 µm sieve. In contrast, the flour with the highest granulometry was the CF, retaining the greatest amount of flour in the 400 and 250 µm sieves. On the other hand, OQC, corresponding to the F2 breader, and BSG presented a PSD similar to that of the CB. This is relevant because particle size can influence the color of the flour, the retention of water and oil, as well as the viscosity of doughs or pastes. It has been found that decreasing the particle size makes flours lighter, increases their luminescence (“L” value, associated with greater contact surface allowing the reflection of light), and promotes hydration properties [26]. Various studies on flours have shown that smaller particle sizes increase the percentages of water absorption and retention [27,28]. This could be related to the greater contact of water with the particles; in addition, with the grinding process, proteins and fibers tend to be more exposed, so they interact and form bonds with water more easily [28]. However, there may be exceptions; Dat [29] found that as particle size increased from 0.18 mm to 0.45 mm in coconut flour, water retention capacity increased, while oil retention capacity decreased with particle size—the same results were reported by Ahmed et al. [26] working with chestnut flour. Finally, particle size influences the baking properties of the mass, affecting parameters of the viscosity profile, and it has been reported that finer powders gelatinize at lower temperatures and show higher viscosity peaks [26].

The F1 breader presented the finest particle size with around 60% passing through the 149 µm sieve (Figure 1B). On the other hand, the F2 breader had a higher percentage of intermediate particles (250 and 170 µm sieves), whereas the CB showed a percentage of retention in the 400 µm sieve similar to that of the F2 breader.

### 3.2. Viscosity Profile Analysis

The viscosity profile analysis showed differences between the flour samples (Figure 2A) and breaders (Figure 2B). The RVA profiles corresponding to WF, OQC, and the CB show similar behavior, developing the maximum peak of viscosity, breakdown, setback, and final viscosity in a more pronounced way than the other samples. The way the first viscosity peak develops reflects how starch granules absorb and bind water, causing physical interactions and consequently, increasing viscosity. This continues until the granules reach their maximum size while being agitated and in contact with other molecules, which causes them to break. The maximum viscosity peak is formed when the rate of swollen granules is equal to that of damaged granules; granules that have a greater swelling power confer higher viscosities [30]. On the other hand, higher peaks are related to higher values of starch in the sample, and the time and temperature to reach these peaks decrease with particle size [26]. This could be because WF is the sample with the highest starch content and the finest flour, followed by the CB and OQC. When the temperature is maintained at 95 °C, the starch granules that have been damaged start leaching amylose, which leads to a decrease in viscosity. In addition, this indicates the stability the sample presents during cooking [26,30].

The CB and the F2 breader (Figure 2B) show greater stability during cooking, since their drop is not as drastic as that of WF. Regarding the cooling stage (from 95 to 50 °C), the samples’ viscosity increases, and we observed that the curves that rise the most are those of WF and F2, followed by the CB and CF. This happens because starch molecules re-associate (retrogradation), forming a gel [30]. Likewise, gels that contain a greater amount of amylopectin will be firmer; however, firmness is also affected by other components of the mixture, such as fat and proteins [31]. In addition, retrogradation has been related to hardening effects in breadcrumbs and an increase in hardness with respect to their texture; however, in foods such as French fries and breading, it is a desirable characteristic [32].

The behavior of the CF during the different stages of the test is quite unstable, which may be due to its composition (high in fat and protein, low in starch). Proteins and lipids interact with the starch, affecting its measured viscosity in the RVA. The interaction between starch and proteins is determined by the nature of the individual proteins. The temperature reduction contributes most to the initial increase in viscosity. As the temperature falls, glucan chains from starch become entangled with each other (retrogradation), forming a gel, which makes a greater contribution to the increase in viscosity as the temperature drops [30]. Regarding the behavior of the samples CPF, SPC, and BSG, it could be attributed to the low levels of starch present [33,34]. It is important to mention that the denaturation of proteins contributes to the increase in viscosity; it has been found that for CF, this occurs at 100–120 °C, which is higher than the maximum temperature of this test [33]. These authors mention that CPF could develop higher viscosity as long as the temperature is increased; therefore, SPC and BSG could present similar situations due to their high-protein content.

Differences in the viscosity profiles of the breaders were observed (Figure 2B). The peak viscosity of CB and F2 were similar (around 1600cP), but CB presented more pronounced shear thinning (breakdown) than the other breaders. F1 showed the lowest viscosity values during the 13 min test with the RVA, which could be related to the contribution of the CPF, SPC, and BSG flours.

### 3.3. Breaders Formulations and Percentage of Adhesion

Table 1 shows the formulations of the two breaders. WF is the one found in the largest proportion of the formulation, followed by CPF due to high WRC values and because it is a color reference in nuggets. Finally, the CF, SPC, and BSG have the lowest percentages; however, protein contribution and water retention are still quite high. The viscosity profile of the F2 breader is similar to that of the CB; this was one criterion for choosing it as a breader, in addition to its chemical composition.

The percentage of “pickup” or adhesion, moisture, and fat contents of the breaded and fried chicken nuggets with CB and those of Formulations 1 and 2 are shown in Table 3. The application of breading agents to the chicken nuggets resulted in “pickup” values of around 3.5%, without significant differences between samples. Brannan et al. [20] obtained an average of 4.5% “pickup” using flour as a breading agent in chicken nuggets, without applying batter, as done in our study. Adhesion values for flour-type breaders remain below 20%. The industry usually regulates coverage percentages in order not to affect the palatability of meat products, keeping them below 30% [35]. Therefore, the formulated breaders presented desirable adhesion characteristics, as they were within the established parameters. The chicken nuggets showed good moisture retention (Table 3), with F1 nuggets having the highest retention (67.6%), which may be related to a higher protein content and its ability to bind water, preventing evaporation [7].

Although F1 showed lower viscosity values (Figure 2B) and thus, an apparent decrease in interaction with water, we observed that raising the temperature to 180 °C during frying prevented moisture migration. This could be related to the unfolding of proteins at high temperatures and the formation of bonds with water and the hydroxyl groups of the fiber [23]. Furthermore, the possible formation of intermolecular disulfide cross-links could have improved the barrier against water removal [13]. Other authors reported a 45% to 65% moisture content when they incorporated ingredients high in fiber and protein in their formulations [20,36,37]. Regarding fat content (Table 3), F1 significantly reduced the fat content during frying, presenting the lowest value (4.5%) and, at the same time, the highest moisture content (67.6%). This could be related to the replacement theory of water [8], which has a direct relationship with the amount of water that evaporates and the oil that penetrates the food during frying. Although F1 presented an intermediate ORC (Table 2) compared to the rest of the samples, this had no relationship during frying; once again, the fiber and the unfolding of the proteins at high temperatures could have beneficial consequences, thus preventing penetration of the oil.

The nuggets covered with F2 had 6.3% fat (Table 3); although the F2 breader showed apparently similar characteristics to the CB in the different tests, the fat content was considerably higher. This could be explained by different factors; one of them is that F2 alone contains 5.5% fat, initially contributing a higher percentage of lipids. The proteins present in the sample, their unfolding during the frying process, and their interaction could have favored the absorption of oil, since the non-polar or hydrophobic amino acids of the proteins tend to bind to the carbohydrate chains and increase oil absorption [38]. In addition to this, the “pickup” of F2 was greater (Table 3), generating a thicker coating, and it is known that when the thickness of the breading in a fried food increases, the oil or fat content also increases [35]. F1 presented the best results regarding fat content, leaving behind the CB and F2, so this formulation shows desirable characteristics for consumers, promoting it as a breader for chicken nuggets [39].

### 3.4. Texture Profile Analysis and Color of Breaded and Fried Chicken Nuggets

The effect of different breaders on the texture of chicken nuggets is presented in Table 4. Nuggets prepared with F1 showed higher values of hardness and chewiness than those with CB and F2. In addition, each texture value of nuggets breaded with F1 coincides with those reported previously [11,13,37]. Even when they had only one layer of the coating system, that is, a fairly thin layer of breading, F1 nuggets showed similar values to those with thicker breading, directly relating to crunchiness [35]. Shan et al. [11] reported hardness values of around 450 g and chewiness of around 225 for nuggets, which are expected values for this type of food. These authors used a covering system made of soy fiber, xanthan gum, and breadcrumbs. Therefore, nuggets with F1, obtaining values of 449.1 g for hardness and 182.7 for chewiness, are comparable even with nuggets composed of several layers and a final breading of greater granulometry. We infer that F1 obtained favorable results thanks to the fact that it could have formed a firm gel through the starch it contained and its high levels of protein and fiber, which, induced by heat, managed to retain moisture and, in turn, give good texture to the nuggets. 

Nuggets with F2 and CB presented a lower chewiness compared to those prepared with F1. Regarding cohesiveness, which explains how well a product supports a second deformation in relation to the resistance to the first deformation, the samples were not different from each other and were similar to those found in the literature [37,40]. Springiness represents how well a product physically recovers after being deformed during the first compression. In this parameter, the chicken nuggets did not present significant differences, agreeing with values previously reported [11,40].

The color of a food is one of the most important attributes for initial acceptance by consumers. F1 nuggets only presented a small increase in L* value (Table 4), which could be related to the flour mix of the breader, since it is made of CPF and CF, which are distinguished by presenting L* values greater than 70 [41,42]. Furthermore, in foods where gels form such as our samples, water binding, dehydration prevention, and Maillard reaction inhibition cause higher luminosity values [35]. The chicken nugget prepared with the F1 breader was significantly lighter than those with the CB and F2 breader, with an L* value close to 55. On the other hand, nuggets prepared with the CB and F1 breader showed similar b* values (around 28), presenting a desirable golden appearance.

The nugget with the F2 breader had a lower yellow tone. F2 nuggets were not different from the control (CB) with respect to the L* value, while a* and b* were lower. Furthermore, F2 nuggets presented a decrease in red and yellow colors, which agrees with a previous report on chicken nuggets where chia was incorporated in the formulation [37]. The characteristics shown by F2 are not desirable in fried foods, since colors in golden and yellow tones are expected [7]. Color measurements of CB and F1 nuggets agree with a previous report [11]; however, other authors report a slight increase in a* values (shades in red), and mention that this could be related to the Maillard and caramelization reactions that occur during the frying process, contributing to color changes [12,13,20].

### 3.5. Proximal Analysis and Dietary Fiber Content of Breaded and Fried Chicken Nuggets

Table 5 shows the results obtained from the proximal analysis of the nuggets prepared with the CB and F1 breader, which highlights a lower fat content in the nuggets made with F1, which is related to the higher moisture content and the water replacement theory, which, in turn, is related to the high levels of dietary fiber and protein [8]. The lipid content is below what was previously reported in the literature, where authors obtained values around 20–25% lipids [11,12,37]. The ash content of the nuggets was not significantly different.

Nuggets with F1 showed a higher TDF value (4.5%, Table 5), exceeding that of the nuggets with the CB by almost double, which is directly related to the composition of the breader agent. This could explain the behavior of the samples with respect to WRC, moisture, and fat content, since F1 showed better results than the CB.

Dietary fiber may have interacted with proteins and contributed to the increases in WRC in the breader, causing less moisture loss during frying and a greater reduction in oil absorption. There are few studies that report a complete characterization of the finished product (chicken nugget), and even fewer that report the total dietary fiber (TDF) content. It has been found that nuggets incorporating 10% chia and chickpea flour contained between 1% and 3% TDF, so F1 nuggets presented higher TDF content than those reported in the literature [36,37]. The daily fiber intake recommendation for adults is at least 25 g [43], so a 100 g portion of nuggets with F1 would provide 18% of the daily requirement, thus increasing their nutritional and commercial value. According to European Parliament and Council legislation [44], the F1-breaded fried chicken nuggets can be claimed as a “source of fiber” since they contain more than 3% fiber.

## 4. Conclusions

Wheat, coconut, chickpeas, and oat-quinoa-chia flours, soy protein concentrate, and brewers’ spent grain showed favorable physicochemical characteristics as ingredients in a breader. The Formulation 1 nuggets have better characteristics than those prepared with the CB since they have reduced fat content and a higher total dietary fiber content. Additionally, they showed a texture profile expected for breaded products, including those with thicker coverage, standing out for their similarity in hardness and chewiness values, which are determining characteristics in fried products. Moreover, the color of the Formulation 1 nuggets was not negatively affected by the flour mixture; on the contrary, it remained within the expected range. Therefore, the Formulation 1 breader made with high-protein and fiber flours improved the nutritional profile of chicken nuggets and maintained the texture and color expected for this type of food. The Formulation 1 breader could be a viable product for the industry, since it stands out from the products currently available on the market for its high content of dietary fiber and protein, offering consumers an option with a greater amount of beneficial nutrients to health. Finally, it adds value to a by-product of the brewing industry and diversifies the use of coconut and chickpea flours.

## Figures and Tables

**Figure 1 foods-12-04463-f001:**
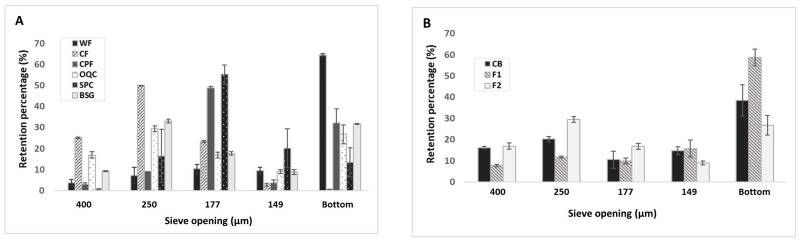
Particle size distribution of the flours (**A**) and breaders. (**B**) WF = Wheat flour; CF = Coconut flour; CPF = Chickpeas flour; SPC = Soy protein concentrate; BSG = Brewers’spent grain, OQC = Oats-quinoa-chia flour; F1: Breader formulation 1 (WF:CF:CPF:SPC:BSG, 50:10:20:10:10); F2: Breader formulation 2 (O:Q:C, 88:8.8:3.2); CB = Commercial breader.

**Figure 2 foods-12-04463-f002:**
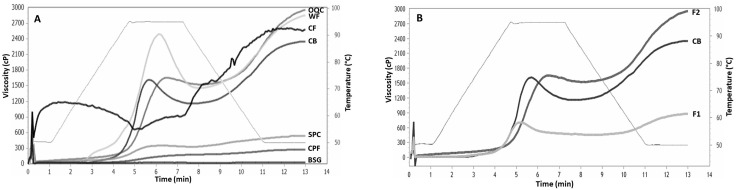
Viscosity profiles or RVA pasting curves of the flours (**A**) and breaders. (**B**) WF = Wheat flour; CF = Coconut flour; CPF = Chickpeas flour; SPC = Soy protein concentrate; BSG = Brewers’spent grain, OQC = Oats-quinoa-chia flour; F1: Breader formulation 1 (WF:CF:CPF:SPC:BSG, 50:10:20:10:10); F2: Breader formulation 2 (O:Q:C, 88:8.8:3.2); CB = Commercial breader.

**Table 1 foods-12-04463-t001:** Formulations used for breading chicken nuggets.

Sample	Formulations (%)	
	F1	F2
Wheat flour	50	0
Coconut flour	10	0
Chickpeas flour	20	0
Soy protein concentrate	10	0
Brewers’ spent grain	10	0
Oats	0	88
Quinoa	0	8.8
Chia	0	3.2

**Table 2 foods-12-04463-t002:** Proximal composition, water retention (WRC), and oil retention capacities (ORCs) of the flour samples and breaders prepared with two formulations (F1 and F2), and a commercial breader (CB) as a control.

Sample	Moisture	Protein	Fat (%)	Ash	WRC	ORC (g/g)
WF	8.0 ± 0.1 ^cd^	14.3 ± 0.1 ^c^	0.9 ± 0.1 ^a^	0.6 ± 0.2 ^a^	63.1 ± 4.9 ^a^	1.83 ± 0.02 ^ab^
CF	10.4 ± 0.4 ^f^	23.1 ± 0.5 ^e^	12.9 ± 0.2 ^f^	3.2 ± 0.0 ^e^	408.0 ± 2.7 ^e^	2.16 ± 0.04 ^e^
CPF	7.5 ± 0.0 ^c^	21.2 ± 0.0 ^d^	6.3 ± 0.5 ^d^	2.6 ± 0.0 ^d^	145.0 ± 2.5 ^b^	1.93 ± 0.02 ^bc^
SPC	5.9 ± 0.2 ^b^	83.3 ± 0.2 ^f^	0.4 ± 0.0 ^a^	3.7 ± 0.0 ^f^	356.2 ± 3.5 ^d^	2.02 ± 0.06 ^d^
BSG	4.8 ± 0.3 ^a^	24.3 ± 0.0 ^e^	8.7 ± 0.0 ^e^	1.7 ± 0.0 ^c^	274.6 ± 0.5 ^c^	2.68 ± 0.02 ^f^
F1	5.1 ± 0.0 ^a^	22.0 ± 0.3 ^c^	4.5 ± 0.3 ^c^	1.8 ± 0.0 ^b^	124.4 ± 0.8 ^b^	1.97 ± 0.01 ^cd^
F2, OQC	9.4 ± 0.5 ^e^	12.6 ± 0.0 ^b^	5.5 ± 0.3 ^c^	1.0 ± 0.0 ^b^	269.5 ± 0.7 ^c^	1.89 ± 0.01 ^abc^
CB	8.6 ± 0.0 ^de^	11.3 ± 0.3 ^a^	1.7 ± 0.1 ^b^	7.7 ± 0.4 ^g^	69.3 ± 6.6 ^a^	1.82 ± 0.03 ^a^

Means with the same superscript letter in a column are not significantly different (*p* > 0.05). WF = Wheat flour; CF = Coconut flour; CPF= Chickpeas flour; SPC = Soy protein concentrate; BSG = Brewers’ spent grain, OQC = Oats-quinoa-chia flour; F1: Breader formulation 1 (WF:CF:CPF:SPC:BSG, 50:10:20:10:10); F2: Breader formulation 2 (O:Q:C, 88:8.8:3.2); CB = Commercial breader.

**Table 3 foods-12-04463-t003:** Percentage of breader pick up, and moisture and fat contents of the breaded and fried chicken nuggets.

Sample	Pick Up (%)	Moisture (%)	Fat (%)
CB	3.6 ± 0.6 ^a^	64.9 ± 0.0 ^a^	5.0 ± 0.4 ^b^
F1	3.1 ± 0.3 ^a^	67.6 ± 0.5 ^b^	4.5 ± 0.1 ^a^
F2	4.0 ± 0.2 ^a^	65.1 ± 0.5 ^a^	6.3 ± 0.4 ^c^

Means with the same superscript letter in a column are not significantly different (*p* > 0.05). CB = Commercial breader; F1 = Breader prepared with Formulation 1; F2 = Breader prepared with Formulation 2.

**Table 4 foods-12-04463-t004:** Texture profile analysis and color measurement values of breaded and fried chicken nuggets.

Sample	Hardness (g)	Cohesiveness	Chewiness	Springiness	L*	a*	b*
CB	379.3 ± 46.8 ^b^	0.5 ± 0.0 ^a^	154.4 ± 24.1 ^b^	0.8 ± 0.1 ^a^	49.1 ± 2.4 ^a^	2.4 ± 0.7 ^b^	27.8 ± 1.2 ^b^
F1	449.1 ± 49.4 ^c^	0.5 ± 0.1 ^a^	182.7 ± 24.1 ^c^	0.9 ± 0.1 ^a^	54.3 ± 0.7 ^b^	3.2 ± 0.3 ^b^	28.5 ± 0.9 ^b^
F2	293.3 ± 34.9 ^a^	0.5 ± 0.0 ^a^	114.2 ± 14.3 ^a^	0.8 ± 0.0 ^a^	51.3 ± 2.8 ^a^	0.0 ± 1.2 ^a^	18.0 ± 1.3 ^a^

Means with the same superscript letter in a column are not significantly different (*p* > 0.05). CB = Commercial breader; F1 = Breader prepared with Formulation 1; F2 = Breader prepared with Formulation 2.

**Table 5 foods-12-04463-t005:** Proximal analysis and dietary fiber content of breaded fried chicken nuggets.

Sample	Moisture (%)	Protein (%)	Fat (%)	Ash (%)	Carbohydrates (%)			
					Total	IDF	SDF	TFD
CB	64.9 ± 0.0 ^a^	23.3 ± 0.2 ^b^	5.0 ± 0.1 ^b^	1.2 ± 0.0 ^a^	5.6 *	2.1 ± 0.5 ^a^	0.2 ± 0.1 ^a^	2.3 ± 0.4 ^a^
F1	67.6 ± 0.5 ^b^	21.9 ± 0.2 ^a^	4.5 ± 0.1 ^a^	1.1 ± 0.0 ^a^	4.9 *	3.9 ± 0.6 ^a^	0.6 ± 0.3 ^a^	4.5 ± 0.9 ^b^

Means with the same superscript letter in a column are not significantly different (*p* > 0.05). CB = Commercial breader; F1 = Breader prepared with the formulation 1. * Calculated by difference.

## Data Availability

The data presented in this study are available within the article.

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
