# Peer review of "Effect of High-Protein and High-Fiber Breaders on Oil Absorption and Quality Attributes in Chicken Nuggets"

_foods, 2023, doi:10.3390/foods12244463_

Round 1
Reviewer 1 Report
Comments and Suggestions for Authors
This study investigated the effect of breaders high in protein and fiber on the oil absorption in chicken nuggets. The manuscript is well written and discuss clearly. Minor revision is suggested:
1. The abstract should include some concrete data.
2. In table1, the fat content of F1 (4.5 ± 0.3b) has no significant different with CB(1.7 ± 0.1b), This does not seem to be accurate, please check the experimental data again and check the results of the statistical analysis.
3. Figure 2 should include the significance.
4. Line 529, “(2018)” should be delete.
5. Please check the reference format carefully, some of the year of publication has not been bolded, such as line460, line 477, line 492.
Reviewer 2 Report
Comments and Suggestions for Authors
The present study analyzed the impact of breaders high in protein and fiber on fat absorption in breaded chicken nuggets. The topic is interesting, and the work offers some useful information. However, the experimental design is too simple and the amount of data is not enough. The writing and terminology should also be thoroughly revised. See comments below.
1. Title. The title description is not very accurate. The results do not show the part of fractions.
2. Abstract. The main question addressed by the research is not clear. Key data is not presented.
3. Introduction. There are various cereals and oilseeds, why do the authors choose Chickpea flour, Coconut flour, Brewers´ spent grain, etc, to study?
4. Methods. The calculation formula of pickup is wrong.
5. Results and discussions.
The contents of moisture, protein, fat, and ash in Table 1 should be presented to the materials sections.
Flours formulations for breaders for nuggets in Table 2 should be presented to the methods sections.
Line 354-356, This sentence is not concise enough, please rephrase it.

The present study analyzed the impact of breaders high in protein and fiber on fat absorption in breaded chicken nuggets. The topic is interesting, and the work offers some useful information. However, the experimental design is too simple and the amount of data is not enough. The writing and terminology should also be thoroughly revised. See comments below.
1. Title. The title description is not very accurate. The results do not show the part of fractions.
2. Abstract. The main question addressed by the research is not clear. Key data is not presented.
3. Introduction. There are various cereals and oilseeds, why do the authors choose Chickpea flour, Coconut flour, Brewers´ spent grain, etc, to study?
4. Methods. The calculation formula of pickup is wrong.
5. Results and discussions.
The contents of moisture, protein, fat, and ash in Table 1 should be presented to the materials sections.
Flours formulations for breaders for nuggets in Table 2 should be presented to the methods sections.
Line 354-356, This sentence is not concise enough, please rephrase it.
Reviewer 3 Report
Comments and Suggestions for Authors
INTRODUCTION quite well written, but there is some information missing on the mechanism of weakening fat sorption during frying depending on the composition of the breading. In the introduction, the authors only mention the composition of the breading used in other works.
materials and methods. I think calling cooked, dried and ground chickpeas "flour" is a mistake. Please develop a more accurate description of this ingredient.
the same mistake is made when specifying soy protein concentrate!!! this is not flour!!
line 102 "and added the amount of water indicated by the equipment software." - ? was it the same for each sample? how much was it exactly? please either provide the range from...g waterg to ...g. Or construct a table and provide quantities.
line 104-105 "During this test, the viscosity of the mixture at different temperatures was recorded," - what does it mean at different temperatures, what was the initial and final temperature, was the viscosity determined at increasing and decreasing temperatures or only at one of the profiles? What was the rate of temperature change, how many degrees per minute.
So what was the origin of "2.4 Breaders formulation and physicochemical properties" line-109. there should be a table here with the exact composition of qualitative quantities.!!!
line 141-142 "To determine the moisture content, the fried chicken nuggets were ground in a Moulinex food processor (Model AR6838)" ...? really, don't you have a laboratory grinder? To increase the reliability of the research, I would omit the name of the home kitchen equipment, but what is important is how long the sample was ground and what weight it was obtained, and what size of sieve it passed through.
Fig. 2. should be the same max values on the axis describing the viscosity changes. I wonder about the meaning of these viscosity measurements where the sample is heated to close to 100 C in 5 minutes?
table 5 why the results for F2 are not presented
I don't understand at all why only two versions of breading were selected, I think it's definitely not enough to be able to publish an article in this form. The origins of the breading mixtures varied greatly, so it is difficult to judge them accurately. I'm wondering what the possible uses are, whether an average consumer has access to a brewery and whether they cook and then dry the chickpeas to make coating. Were the dried chickpeas also ground in a Mulinex food processor?
Reviewer 4 Report
Comments and Suggestions for Authors
The present manuscript evaluated the effect of two breaders high in protein and fiber on the oil absorption and quality of chicken nuggets. The manuscript has some important recommendations. The hypothesis is sound. The language is clear and easy to understand.
Title: May add quality attributes also
Abstract: More clarity on breeders is needed (with their protein and fiber content) for better understanding and improving the readership impact of the manuscript. Then readers will understand the effect of high protein and fiber on oil absorption during frying nuggets. Further, please indicate p levels also.
Keyword: add nutrition quality also as keyword.
Introduction: Well-described the background information and justified the need to take this study.
Section 2.4: need detailing or state mention in the section section 3.3 and Table 2.
Section 2.8: Please mention sample size
Proximate composition of F2 samples was not performed?.
Conclusion: Appropriate
Round 2
Reviewer 3 Report
Comments and Suggestions for Authors
I see significant improvement in the way results are presented and discussed.
Author Response
Thanks for your comment and revision.